# Consistent Feature Selection for Analytic Deep Neural Networks

**Vu Dinh** *
Department of Mathematical Sciences
University of Delaware
Delaware, USA
vucdinh@udel.edu

**Lam Si Tung Ho** *
Department of Mathematics and Statistics
Dalhousie University
Halifax, Nova Scotia, Canada
Lam.Ho@dal.ca

## Abstract

One of the most important steps toward interpretability and explainability of neural network models is feature selection, which aims to identify the subset of relevant features. Theoretical results in the field have mostly focused on the prediction aspect of the problem with virtually no work on feature selection consistency for deep neural networks due to the model's severe nonlinearity and unidentifiability. This lack of theoretical foundation casts doubt on the applicability of deep learning to contexts where correct interpretations of the features play a central role.

In this work, we investigate the problem of feature selection for analytic deep networks. We prove that for a wide class of networks, including deep feed-forward neural networks, convolutional neural networks, and a major sub-class of residual neural networks, the Adaptive Group Lasso selection procedure with Group Lasso as the base estimator is selection-consistent. The work provides further evidence that Group Lasso might be inefficient for feature selection with neural networks and advocates the use of Adaptive Group Lasso over the popular Group Lasso.

## 1 Introduction

In recent years, neural networks have become one of the most popular models for learning systems for their strong approximation properties and superior predictive performance. Despite their successes, their "black box" nature provides little insight into how predictions are being made. This lack of interpretability and explainability stems from the fact that there has been little work that investigates statistical properties of neural networks due to its severe nonlinearity and unidentifiability.

One of the most important steps toward model interpretability is feature selection, which aims to identify the subset of relevant features with respect to an outcome. Numerous techniques have been developed in the statistics literature and adapted to neural networks over the years. For neural networks, standard Lasso is not ideal because a feature can only be dropped if all of its connections have been shrunk to zero together, an objective that Lasso does not actively pursue. Zhao et al. [2015], Scardapane et al. [2017] and Zhang et al. [2019] address this concern by utilizing Group Lasso and its variations for selecting features of deep neural networks (DNNs). Feng and Simon [2017] propose fitting a neural network with a Sparse Group Lasso penalty on the first-layer input weights while adding a sparsity penalty to later layers. Li et al. [2016] propose adding a sparse one-to-one linear layer between the input layer and the first hidden layer of a neural network and performing $\ell_1$ regularization on the weights of this extra layer. Similar ideas are also applied to various types of networks and learning contexts [Tank et al., 2018, Ainsworth et al., 2018, Lemhadri et al., 2019].

Beyond regularization-based methods, Horel and Giesecke [2019] develop a test to assess the statistical significance of the feature variables of a neural network. Alternatively, a ranking of feature

---

importance based on local perturbations, which includes fitting a model in the local region around the input or locally perturbing the input to see how predictions change, can also be used as a proxy for feature selection [Simonyan et al., 2013, Ibrahim et al., 2014, Ribeiro et al., 2016, Nezhad et al., 2016, Lundberg and Lee, 2017, Shrikumar et al., 2017, Ching et al., 2018, Taherkhani et al., 2018, Lu et al., 2018]. "Though these methods can yield insightful interpretations, they focus on specific architectures of DNNs and can be difficult to generalize" [Lu et al., 2018].

Despite the success of these approaches, little is known about their theoretical properties. Results in the field have been either about shallow networks with one hidden layer [Dinh and Ho, 2020, Liang et al., 2018, Ye and Sun, 2018], or focus on posterior concentration, prediction consistency, parameter-estimation consistency and convergence of feature importance [Feng and Simon, 2017, Farrell et al., 2018, Polson and Ročková, 2018, Fallahgoul et al., 2019, Shen et al., 2019, Liu, 2019], with virtually no work on feature selection consistency for deep networks. This lack of a theoretical foundation for feature selection casts doubt on the applicability of deep learning to applications where correct interpretations play a central role such as medical and engineering sciences. This is problematic since works in other contexts indicated that Lasso and Group Lasso could be inconsistent/inefficient for feature selection [Zou, 2006, Zhao and Yu, 2006, Wang and Leng, 2008], especially when the model is highly-nonlinear [Zhang et al., 2018].

In this work, we investigate the problem of feature selection for deep networks. We prove that for a wide class of deep analytic neural networks, the GroupLasso + AdaptiveGroupLasso procedure (i.e., the Adaptive Group Lasso selection with Group Lasso as the base estimator) is feature-selection-consistent. The results also provide further evidence that Group Lasso might be inconsistent for feature selection and advocate the use of the Adaptive Group Lasso over the popular Group Lasso.

## 2 Feature selection with analytic deep neural networks

Throughout the paper, we consider a general analytic neural network model described as follows. Given an input $x$ that belongs to be a bounded open set $\mathcal{X} \subset \mathbf{R}^{d_0}$, the output map $f_\alpha(x)$ of an $L$-layer neural network with parameters $\alpha = (P, p, S, Q, q)$ is defined by

- input layer: $h_1(x) = P \cdot x + p$
- hidden layers: $h_j(x) = \phi_{j-1}(S, h_{j-1}(x), h_{j-2}(x), \ldots, h_1(x)), \ \ j = 2, \ldots, L-1.$
- output layer: $f_\alpha(x) = h_L(x) = Q \cdot h_{L-1}(x) + q$

where $d_i$ denotes the number of nodes in the $i$-th hidden layer, $P \in \mathbf{R}^{d_1 \times d_0}$, $Q \in \mathbf{R}^{d_L \times d_{L-1}}$, $p \in \mathbf{R}^{d_1}$, $q \in \mathbf{R}^{d_L}$, and $\phi_1, \phi_2, \ldots, \phi_{L-2}$ are analytic functions parameterized by the hidden layers' parameter $S$. This framework allows interactions across layers of the network architecture and only requires that (i) the model interacts with inputs through a finite set of linear units, and (ii) the activation functions are analytic. This parameterization encompasses a wide class of models, including feed-forward networks, convolutional networks, and (a major subclass of) residual networks.

Hereafter, we assume that the set of all feasible vectors $\alpha$ of the model is a hypercube $\mathcal{W} = [-A, A]^{n_\alpha}$ and use the notation $h^{[i]}$ to denote the $i$-th component of a vector $h$ and $u^{[i,k]}$ to denote the $[i,k]$-entry of a matrix $u$. We study the feature selection problem for regression in the model-based setting:

**Assumption 2.1.** *Training data $\{(X_i, Y_i)\}_{i=1}^n$ are independent and identically distributed (i.i.d ) samples generated from $P^*_{X,Y}$ such that the input density $p_X$ is positive and continuous on its domain $\mathcal{X}$ and $Y_i = f_{\alpha^*}(X_i) + \epsilon_i$ where $\epsilon_i \sim \mathcal{N}(0, \sigma^2)$ and $\alpha^* \in \mathcal{W}$.*

**Remarks.**   Although the analyses of the paper focus on Gaussian noise, the results also apply to all models for which $\epsilon^2$ is sub-Gaussian, including the cases of bounded noise. The assumption that the input density $p_X$ is positive and continuous on its bounded domain $\mathcal{X}$ ensures that there is no perfect correlations among the inputs and plays an important role in our analysis. Throughout the paper, the network is assumed to be fixed, and we are interested in the asymptotic behaviors of estimators in the learning setting when the sample size $n$ increases.

We assume that the "true" model $f_{\alpha^*}(x)$ only depends on $x$ through a subset of significant features while being independent of the others. The goal of feature selection is to identify this set of significant features from the given data. For convenience, we separate the inputs into two groups $s \in \mathbb{R}^{n_s}$ and

$z \in \mathbb{R}^{n_z}$ (with $n_s + n_z = d_0$) that denote the significant and non-significant variables of $f_{\alpha^*}(x)$, respectively. Similarly, the parameters of the first layer ($P$ and $p$) are grouped by their corresponding input, i.e.

$$x = (s, z), \qquad P = (u, v) \qquad \text{and} \qquad p = (b_1, b_2)$$

where $u \in \mathbb{R}^{d_1 \times n_s}$, $v \in \mathbb{R}^{d_1 \times n_z}$, $b_1 \in \mathbb{R}^{n_s}$ and $b_2 \in \mathbb{R}^{n_z}$. We note that this separation is simply for mathematical convenience and the training algorithm is not aware of such dichotomy.

We define significance and selection consistency as follows.

**Definition 2.2** (Significance). *The $i$-th input $x^{[i]}$ of a network $f_\alpha(x)$ is referred to as non-significant iff the output of the network does not depend on the value of that input variable. That is, let $g_i(x, s)$ denote the vector obtained from $x$ by replacing the $i$-th component of $x$ by $s \in \mathbb{R}$, we have $f_\alpha(x) = f_\alpha(g_i(x, s))$ for all $x \in \mathcal{X}, s \in \mathbb{R}$.*

**Definition 2.3** (Feature selection consistency). *An estimator $\alpha_n$ with first layer's parameters $(u_n, v_n)$ is feature selection consistent if for any $\delta > 0$, there exists $N_\delta$ such that for $n > N_\delta$, we have*

$$u_n^{[:,k]} \neq 0, \ \forall k = 1, \ldots, n_s, \qquad \text{and} \qquad v_n^{[:,l]} = 0, \ \forall l = 1, \ldots, n_z$$

*with probability at least $1 - \delta$.*

One popular method for feature selection with neural networks is the Group Lasso (GL). Moreover, since the model of our framework only interacts with the inputs through the first layer, it is reasonable that the penalty should be imposed only on these parameters. A simple GL estimator for neural networks is thus defined by

$$\hat{\alpha}_n := \underset{\alpha=(u,v,b_1,b_2,S,Q,q)}{\operatorname{argmin}} \frac{1}{n} \sum_{i=1}^{n} \ell(\alpha, X_i, Y_i) + \lambda_n L(\alpha) \ \text{ where } \ L(\alpha) = \sum_{k=1}^{n_s} \|u^{[:,k]}\| + \sum_{l=1}^{n_z} \|v^{[:,l]}\|,$$

$\ell(\alpha, x, y) = (y - f_\alpha(x))^2$ is the square-loss, $\lambda_n > 0$, $\|\cdot\|$ is the standard Euclidean norm and $u^{[:,k]}$ is the vector of parameters associated with $k$-th significant input.

While GL and its variation, the Sparse Group Lasso, have become the foundation for many feature selection algorithms in the field, there is no known result about selection consistency for this class of estimators. Furthermore, like the regular Lasso, GL penalizes groups of parameters with the same regularization strengths and it has been shown that excessive penalty applied to the significant variables can affect selection consistency [Zou, 2006, Wang and Leng, 2008]. To address these issues, Dinh and Ho [2020] propose the "GroupLasso+AdaptiveGroupLasso" (GL+AGL) estimator for feature selection with neural networks, defined as

$$\tilde{\alpha}_n := \underset{\alpha=(u,v,b_1,b_2,S,Q,q)}{\operatorname{argmin}} \frac{1}{n} \sum_{i=1}^{n} \ell(\alpha, X_i, Y_i) + \zeta_n M_n(\alpha),$$

where

$$M_n(\alpha) = \sum_{k=1}^{n_s} \frac{1}{\|\hat{u}_n^{[:,k]}\|^\gamma} \|u^{[:,k]}\| + \sum_{k=1}^{n_z} \frac{1}{\|\hat{v}_n^{[:,k]}\|^\gamma} \|v^{[:,k]}\|.$$

Here, we use the convention $0/0 = 1$, $\gamma > 0$, $\zeta_n$ is the regularizing constant and $\hat{u}_n$, $\hat{v}_n$ denotes the $u$ and $v$ components of the GL estimate $\hat{\alpha}_n$. As typical with adaptive lasso estimators, GL+AGL uses its base estimator to provide a rough data-dependent estimate to shrink groups of parameters with different regularization strengths. As $n$ grows, the weights for non-significant features get inflated (to infinity) while the weights for significant ones remain bounded [Zou, 2006]. Dinh and Ho [2020] prove that GL+AGL is selection-consistent for irreducible shallow networks with hyperbolic tangent activation. In this work, we argue that the results can be extended to general analytic deep networks.

## 3 Consistent feature selection via GroupLasso+AdaptiveGroupLasso

The analysis of GL+AGL is decomposed into two steps. First, we establish that the Group Lasso provides a good proxy to estimate regularization strengths. Specifically, we show that

- The $u$-components of $\hat{\alpha}_n$ are bounded away from zero

- The $v$-components of $\hat{\alpha}_n$ converge to zero with a polynomial rate.

Second, we prove that a selection procedure based on the weights obtained from the first step can correctly select the set of significant variables given informative data with high probability.

One technical difficulty of the proof concerns with the geometry of the risk function around the set of risk minimizers

$$\mathcal{H}^* = \{\alpha : R(\alpha) = R(\alpha^*)\}$$

where $R(\alpha)$ denotes the risk function $R(\alpha) = \mathbb{E}_{(X,Y)\sim P^*_{X,Y}}[(f_\alpha(X) - Y)^2]$. Since deep neural networks are highly unidentifiable, the set $\mathcal{H}^*$ can be quite complex. For example:

(i) A simple rearrangement of the nodes in the same hidden layer leads to a new configuration that produces the same mapping as the generating network

(ii) If either all in-coming weights or all out-coming weights of a node is zero, changing the others also has no effect on the output

(iii) For hyperbolic tangent activation, multiplying all the in-coming and out-coming parameters associated with a node by -1 also lead to an equivalent configuration

These unidentifiability create a barrier in studying deep neural networks. Existing results in the field either are about equivalent graph transformation [Chen et al., 1993] or only hold in some generic sense [Fefferman and Markel, 1994] and thus are not sufficient to establish consistency. Traditional convergence analyses also often rely on local expansions of the risk function around isolated optima, which is no longer the case for neural networks since $H^*$ may contain subsets of high dimension (e.g., case (ii) above) and the Hessian matrix (and high-order derivatives) at an optimum might be singular.

In this section, we illustrate that these issues can be avoided with two adjustments. First, we show that while the behavior of a generic estimator may be erratic, the regularization effect of Group Lasso constraits it to converge to a subset $\mathcal{K} \subset \mathcal{H}$ of "well-behaved" optima. Second, instead of using local expansions, we employ Lojasewicz's inequality for analytic functions to upper bound the distance from $\hat{\alpha}_n$ to $\mathcal{H}^*$ by the excess risk. This removes the necessity of the regularity of the Hessian matrix and enlarges the class of networks for which selection consistency can be analyzed.

## 3.1 Characterizing the set of risk minimizers

**Lemma 3.1.**      *(i) There exists $c_0 > 0$ such that $\|u_\alpha^{[:,k]}\| \geq c_0$ for all $\alpha \in \mathcal{H}^*$ and $k = 1, \ldots, n_s$.*

*(ii) For $\alpha \in \mathcal{H}^*$, the vector $\phi(\alpha)$, obtained from $\alpha$ be setting its $v$-components to zero, also belongs to $\mathcal{H}^*$.*

*Proof.* We first prove that $\alpha_0 \in \mathcal{H}^*$ if and only if $f_{\alpha_0} = f_{\alpha^*}$. Indeed, since $f_{\alpha^*}(X) = E_{P^*_{X,Y}}[Y|X]$, we have

$$R(\alpha^*) = \min_g \mathbb{E}_{Y\sim P^*_{X,Y}}[(g(X) - Y)^2] \leq \min_{\alpha \in \mathcal{W}} R(\alpha) = R(\alpha_0)$$

with equality happens only if $f_{\alpha_0} = f_{\alpha^*}$ a.s. on the support of $p_X(x)$. Since $p_X(x)$ is continuous and positive on its open domain $\mathcal{X}$ and the maps $f_\alpha$ are analytic, we deduce that $f_{\alpha_0} = f_{\alpha^*}$ everywhere.

(i) Assuming that no such $c_0$ exists, since $\mathcal{W}$ is compact and $f_\alpha$ is an analytic function in $\alpha$ and $x$, we deduce that there exists $\alpha_0 \in \mathcal{H}^*$ and $k$ such that $u_{\alpha_0}^{[:,k]} = 0$. This means $f_{\alpha_0} = f_{\alpha^*}$ does not depend on significant input $s_k$, which is a contradiction. (ii) Since $\alpha \in \mathcal{H}^*$, we have $f_{\alpha^*}(s,z) = f_\alpha(s,z) = f_\alpha(s,0) = f_{\phi(\alpha)}(s,z)$, which implies $\phi(\alpha) \in \mathcal{H}^*$.     $\square$

**Remarks.**    Lemma 3.1 provides a way to study the behaviors of Group Lasso without a full characterization of the geometry of $\mathcal{H}^*$ as in Dinh and Ho [2020]. First, as long as $d(\hat{\alpha}_n, \mathcal{H}^*) \to 0$, it is straight forward that its $u$-components are bounded away from zero (part (i)). Second, it shows that for all $\alpha \in \mathcal{H}^*$, $\phi(\alpha)$ is a "better" hypothesis in terms of penalty while remaining an optimal hypothesis in terms of prediction. This enables us to prove that if we define

$$\mathcal{K} = \{\alpha \in \mathcal{W} : f_\alpha = f_{\alpha^*} \text{ and } v_\alpha = 0\},$$

and $\lambda_n$ converges to zero slowly enough, the regularization term will force $d(\hat{\alpha}_n, \mathcal{K}) \to 0$. This helps establish that Group Lasso provides a good proxy to construct appropriate regularization strengths.

To provide a polynomial convergence rate of $\hat{\alpha}_n$, we need the following Lemma.

**Lemma 3.2.** *There exist $c_2, \nu > 0$ and such that $R(\beta) - R(\alpha^*) \geq c_2 d(\beta, \mathcal{H}^*)^\nu$ for all $\beta \in \mathcal{W}$.*

*Proof.* We first note that since $f_\alpha(x)$ is analytic in both $\alpha$ and $x$, the excess risk $g(\alpha) = R(\alpha) - R(\alpha^*)$ is also analytic in $\alpha$. Thus $\mathcal{H}^* = \{\alpha : R(\alpha) = R(\alpha^*)\}$ is the zero level-set of the analytic function $g$. By Lojasewicz's inequality for algebraic varieties [Ji et al., 1992], there exist positive constants $C$ and $\nu$ such that $d(\beta, \mathcal{H}^*)^\nu \leq C|g(\beta)| \quad \forall \beta \in \mathcal{W}$, which completes the proof. $\qquad\square$

We note that for cases when $\mathcal{H}^*$ is finite, Lemma 3.2 reduces to the standard Taylor's inequality around a local optimum, with $\nu = 2$ if the Hessian matrix at the optimum is non-singular [Dinh and Ho, 2020]. When $\mathcal{H}^*$ is a high-dimensional algebraic set, Lojasewicz's inequality and Lemma 3.2 are more appropriate. For example, for $g(x, y) = (x - y)^2$, it is impossible to attain an inequality of the form

$$g(x', y') - g(x_0, x_0) \geq C \cdot d((x', y'), (x_0, x_0))^\nu$$

in any neighborhood of any minimum $(x_0, x_0)$, while it is straightforward that

$$g(x', y') - g(x_0, x_0) \geq d((x', y'), Z)^2 \quad \text{where} \quad Z = \{(x, x) : x \in \mathbb{R}\}.$$

This approach provides a way to avoid dealing with model unidentfiability but requires some adaptation of the analysis to accommodate a new mode of convergence.

## 3.2 Convergence of Group Lasso

The two Lemmas in the previous section enables us to analyze the convergence of the Group Lasso estimate, in the sense that $d(\hat{\alpha}_n, \mathcal{K}) \to 0$. First, we define the empirical risk function

$$R_n(\alpha) = \frac{1}{n} \sum_{i=1}^{n} (f_\alpha(X_i) - Y_i)^2$$

and note that since the network of our framework is fixed, the learning problem is continuous and parametric, for which a standard generalization bound as follows can be obtained (proof in Appendix).

**Lemma 3.3** (Generalization bound). *For any $\delta > 0$, there exist $c_1(\delta) > 0$ such that*

$$|R_n(\alpha) - R(\alpha)| \leq c_1 \frac{\log n}{\sqrt{n}}, \qquad \forall \alpha \in \mathcal{W}$$

*with probability at least $1 - \delta$.*

Combining Lemmas 3.2 and 3.3, we have

**Theorem 3.4** (Convergence of Group Lasso). *For any $\delta > 0$, there exist $C_\delta, C' > 0$ and $N_\delta > 0$ such that for all $n \geq N_\delta$,*

$$d(\hat{\alpha}_n, \mathcal{H}^*) \leq C_\delta \left( \lambda_n^{\nu/(\nu-1)} + \frac{\log n}{\sqrt{n}} \right)^{1/\nu} \quad \text{and} \quad \|\hat{v}_n\| \leq 4c_1 \frac{\log n}{\lambda_n \sqrt{n}} + C' \, d(\hat{\alpha}_n, \mathcal{H}^*)$$

*with probability at least $1 - \delta$. Moreover, if $\lambda_n \sim n^{-1/4}$, then with probability at least $1 - \delta$,*

$$d(\hat{\alpha}_n, \mathcal{H}^*) \leq C \left( \frac{\log n}{n} \right)^{\frac{1}{4(\nu-1)}} \quad \text{and} \quad \|\hat{v}_n\| \leq C \left( \frac{\log n}{n} \right)^{\frac{1}{4(\nu-1)}}.$$

*Proof.* Let $\phi(\alpha)$ denote the weight vector obtained from $\alpha$ by setting the $v$-components to zero. If we define $\beta_n = \operatorname{argmin}_{\alpha \in \mathcal{H}^*} \|\hat{\alpha}_n - \alpha\|$ then $\phi(\beta_n) \in \mathcal{H}^*$ and $R(\beta_n) = R(\phi(\beta_n))$.

Since $L(\alpha)$ is a Lipschitz function, we have

$$c_2 d(\hat{\alpha}_n, \mathcal{H}^*)^\nu = c_2 \|\beta_n - \hat{\alpha}_n\|^\nu \leq R(\hat{\alpha}_n) - R(\beta_n)$$

$$\leq 2c_1 \frac{\log n}{\sqrt{n}} + \lambda_n \left( L(\beta_n) - L(\hat{\alpha}_n) \right) \leq 2c_1 \frac{\log n}{\sqrt{n}} + \lambda_n C \|\beta_n - \hat{\alpha}_n\|$$

which implies (through Young's inequality, details in Appendix) that

$$\|\beta_n - \hat{\alpha}_n\|^\nu \le C_1 \lambda_n^{\nu/(\nu-1)} + C_2 \frac{\log n}{\sqrt{n}}.$$

Let $K$ denote the part of the regularization term without the $v$-component. We note that $K$ is a Lipschitz function, $K(\phi(\alpha)) = K(\alpha)$ for all $\alpha$, and $R(\phi(\beta_n)) = R(\hat{\alpha}_n)$. Thus,

$$\begin{aligned}
\lambda_n \sum_l \|\hat{v}_n^{[:,l]}\| &\le R_n(\phi(\beta_n)) - R_n(\hat{\alpha}_n) + \lambda_n[K(\phi(\beta_n)) - K(\hat{\alpha}_n)] \\
&\le 2c_1 \frac{\log n}{\sqrt{n}} + R(\phi(\beta_n)) - R(\hat{\alpha}_n) + \lambda_n[K(\beta_n) - K(\hat{\alpha}_n)] \\
&\le 2c_1 \frac{\log n}{\sqrt{n}} + \lambda_n C \|\beta_n - \hat{\alpha}_n\|.
\end{aligned}$$

This completes the proof. $\qquad\square$

Together, the two parts of Theorem 3.4 shows that the Group Lasso estimator converges to the set of "well-behaved" optimal hypotheses $\mathcal{K} = \{\alpha \in \mathcal{W} : f_\alpha = f_{\alpha^*} \text{ and } v_\alpha = 0\}$ (proof in Appendix)..

**Corollary 3.5.** *For any $\delta > 0$, there exist $C_\delta > 0$ and $N_\delta > 0$ such that for all $n \ge N_\delta$,*

$$\mathbb{P}\left[ d(\hat{\alpha}_n, \mathcal{K}) \le 4c_1 \frac{\log n}{\lambda_n \sqrt{n}} + C_\delta \left( \lambda_n^{\nu/(\nu-1)} + \frac{\log n}{\sqrt{n}} \right)^{1/\nu} \right] \ge 1 - \delta.$$

### 3.3 Feature selection consistency of GroupLasso + AdaptiveGroupLasso

We are now ready to prove the main theorem of our paper.

**Theorem 3.6** (Feature selection consistency of GL+AGL). *Let $\gamma > 0$, $\epsilon > 0$, $\lambda_n \sim n^{-1/4}$, and $\zeta_n = \Omega(n^{-\gamma/(4\nu-4)+\epsilon})$, then the GroupLasso+AdaptiveGroupLasso is feature selection consistent.*

*Proof.* Since $d(\hat{\alpha}_n, \mathcal{H}^*) \to 0$ (Theorem 3.4), we have $\min_{\alpha \in \mathcal{H}^*} \|\hat{u}_n^{[:,k]} - u^{[:,k]}\| \to 0$ for all $k$. By Lemma 3.1, we conclude that $\hat{u}_n^{[:,k]}$ is bounded away from zero as $n \to \infty$. Thus,

$$M_n(\alpha^*) = \sum_{i=1}^{n_s} \frac{1}{\|\hat{u}_n\|^\gamma} \|u^{*[:,k]}\| < \infty \qquad \text{and}$$

$$c_2 d(\tilde{\alpha}_n, \mathcal{H}^*)^\nu \le R(\tilde{\alpha}_n) - R(\alpha^*) \le 2c_1 \frac{\log n}{\sqrt{n}} + \zeta_n \left( M_n(\alpha^*) - M_n(\hat{\alpha}_n) \right) \le 2c_1 \frac{\log n}{\sqrt{n}} + \zeta_n M_n(\alpha^*)$$

which shows that $d(\tilde{\alpha}_n, \mathcal{H}^*) \to 0$. Thus $\tilde{u}_n^{[:,k]}$ is also bounded away from zero for $n$ large enough.

We now assume that $\tilde{v}_n^{[:,k]} \ne 0$ for some $k$ and define a new weight configuration $g_n$ obtained from $\tilde{\alpha}_n$ by setting the $v^{[:,k]}$ component to 0. By definition of the estimator $\tilde{\alpha}_n$, we have

$$R_n(\tilde{\alpha}_n) + \zeta_n \frac{1}{\|\hat{v}_n^{[:,k]}\|^\gamma} \|\tilde{v}_n^{[:,k]}\| \le R_n(g_n).$$

By the (probabilistic) Lipschitzness of the empirical risk (proof in Appendix), there exists $M_\delta$ s.t.

$$\zeta_n \frac{1}{\|\hat{v}_n^{[:,k]}\|^\gamma} \|\tilde{v}_n^{[:,k]}\| \le R_n(g_n) - R_n(\tilde{\alpha}_n) \le M_\delta \|g_n - \tilde{\alpha}_n\| = M_\delta \|\tilde{v}_n^{[:,k]}\|$$

with probability at least $1 - \delta$. Since $\tilde{v}_n^{[:,k]} \ne 0$, we deduce that $\zeta_n \frac{1}{\|\hat{v}_n^{[:,k]}\|^\gamma} \le M_\delta$. This contradicts Theorem 3.4, which proves that for $n$ large enough

$$\zeta_n \frac{1}{\|\hat{v}_n^{[:,k]}\|^\gamma} \ge C_\delta^{-\gamma} \zeta_n \left( \frac{n}{\log n} \right)^{\frac{\gamma}{4(\nu-1)}} \ge 2M_\delta$$

with probability at least $1 - \delta$. This completes the proof. $\qquad\square$

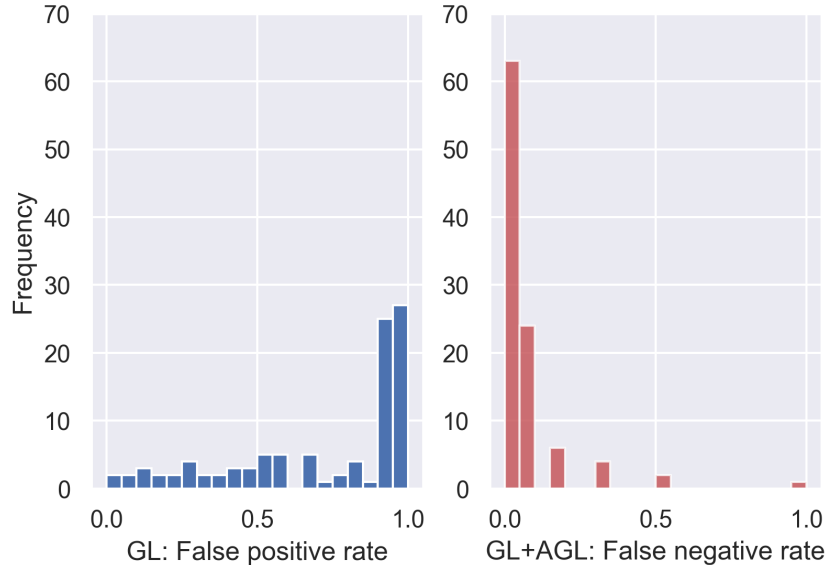

Figure 1: The false positive rate and false negative rate of GL (left panel) and GL+AGL (right panel), respectively. Note that the true positive rate of GL and true negative rate of GL+AGL are 100% for all but one run and are not shown in the figure.

**Remark.** One notable aspect of Theorem 3.6 is the absence of regularity conditions on the correlation of the inputs often used in lasso-type analyses, such as Irrepresentable Conditions [Meinshausen and Bühlmann, 2006] and [Zhao and Yu, 2006], restricted isometry property [Candes and Tao, 2005], restricted eigenvalue conditions [Bickel et al., 2009, Meinshausen and Yu, 2009] or sparse Riesz condition [Zhang and Huang, 2008]. We recall that by using different regularization strengths for individual parameters, adaptive lasso estimators often require less strict conditions for selection consistency than standard lasso. For linear models, the classical adaptive lasso only assumes that the limiting design matrix is positive definite, i.e., there is no perfect correlation among the inputs) [Zou, 2006]. In our framework, this corresponds to the assumption that the density of $X$ is positive on its open domain (Assumption 2.1), which plays an essential role in the proof of Lemma 3.1.

### 3.4 Simulations

To further illustrate the theoretical findings of the paper, we use both synthetic and real data to investigate algorithmic behaviors of GL and GL+AGL. [2] The simulations, implemented in Pytorch, focus on single-output deep feed-forward networks with three hidden layers of constant width. In these experiments, regularizing constants are chosen from a course grid $\{0.001, 0.01, 0.05, 0.1, 0.5, 1, 2\}$ with $\gamma = 2$ using average test errors from random train-test splits of the corresponding dataset. The algorithms are trained over 20000 epochs using proximal gradient descent, which allows us to identify the exact support of estimators without having to use a cut-off value for selection.

In the first experiment, we consider a network with three hidden layers of 20 nodes. The input consists of 50 features, 10 of which are significant while the others are rendered insignificant by setting the corresponding weights to zero. We generate 100 datasets of size $n = 5000$ from the generic model $Y = f_{\alpha^*}(X) + \epsilon$ where $\epsilon \sim \mathcal{N}(0, 1)$ and non-zero weights of $\alpha^*$ are sampled independently from $\mathcal{N}(0, 1)$. We perform GL and GL+AGL on each simulated dataset with regularizing constants chosen using average test errors from three random three-fold train-test splits. We observe that *overall, GL+AGL have a superior performance, selecting the correct support in 63 out of 100 runs, while GL cannot identify the support in any run*. Except for one pathological case when both GL and GL+AGL choose a constant model, GL always selects the correct significant inputs but fail to de-select the

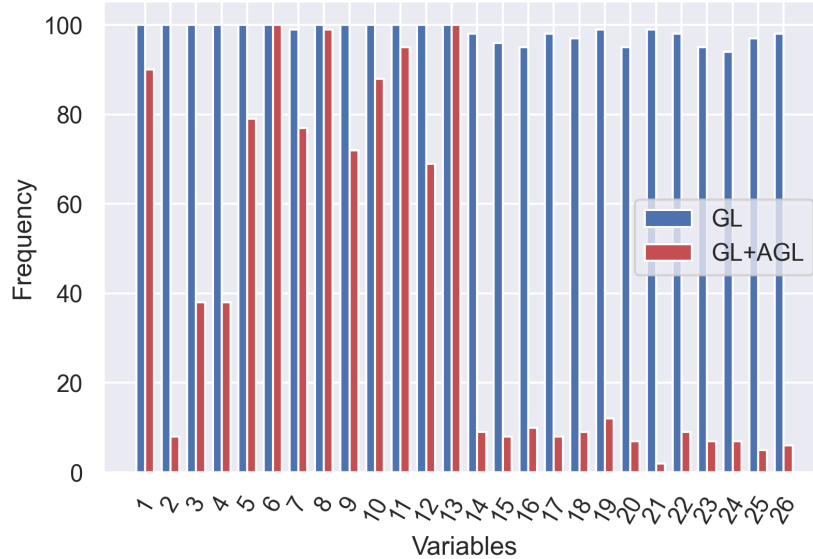

Figure 2: Performance on the Boston Housing dataset: The frequency of being selected by GL (blue) and GL+AGL (red) of each of the 26 predictors. Note that the predictors $14 - 26$ are random Gaussian noise predictors while $1 - 13$ are from the original dataset.

insignificant ones (Figure 1, left panel) while GL+AGL always performs well with the insignificant inputs but sometimes over-shrinks the significant ones (Figure 1, right panel).

Next, we apply the methods to the Boston housing dataset [3]. This dataset consists of 506 observations of house prices and 13 predictors. To analyze the data, we consider a network with three hidden layers of 10 nodes. GL and GL+AGL are then performed on this dataset using average test errors from 20 random train-test splits (with the size of the test sets being $25\%$ of the original dataset). GL identifies all 13 predictors as important, while GL+AGL only selects 11 of them. To further investigate the robustness of the results, we follow the approach of Lemhadri et al. [2019] to add 13 random Gaussian noise predictors to the original dataset for analysis. 100 such datasets are created to compare the performance of GL against GL+AGL using the same experimental setting as above. The results are presented in Figure 2, for which we observe that GL struggles to distinguish the random noises from the correct predictors. We note that Lemhadri et al. [2019] identifies 11 of the original predictors along with 2 random predictors as the optimal set of features for prediction, which is consistent with the performance of GL+AGL.

## 4   Conclusions and Discussions

In this work, we prove that GL+AGL is feature-selection-consistent for all analytic deep networks that interact with inputs through a finite set of linear units. Both theoretical and simulation results of the work advocate the use of GL+AGL over the popular Group Lasso for feature selection.

### 4.1   Comparison to related works

To the best of our knowledge, this is the first work that establishes selection consistency for deep networks. This is in contrast to Dinh and Ho [2020], Liang et al. [2018] and Ye and Sun [2018], which only provide results for shallow networks with one hidden layer, or Polson and Ročková [2018], Feng and Simon [2017], Liu [2019], which focus on posterior concentration, prediction consistency, parameter-estimation consistency and convergence of feature importance. We note that for classical linear model with lasso estimate, it is known that (see Section 2 of Zou [2006] and the discussion therein) the lasso estimator is parameter-estimation consistent as long as the

regularizing parameter $\lambda_n \to 0$ (Zou [2006], Lemmas 2 and 3), but is not feature-selection consistent for $\lambda_n \sim n^{-1/2}$ (Zou [2006], Proposition 1) or for all choices of $\lambda_n$ if some necessary condition on the covariance matrix is not satisfied (Zou [2006], Theorem 1). For both linear model and neural network, parameter-estimation consistency directly implies prediction consistency and convergence of feature importance. In general, there's no known trivial way to extending these approaches to obtain feature-selection-consistency.

Moreover, existing works usually assume that the network used for training has exactly the same size as the minimal network that generates the data. This is assumption is either made explicitly (as in Dinh and Ho [2020]) or implicitly implied by the regularity of the Hessian matrix at the optima (Assumption 5 in Ye and Sun [2018] and Condition 1 in Feng and Simon [2017]). We note that these latter two conditions cannot be satisfied if the size of the training network is not "correct" (for example, when data is generated by a one-hidden-layer network with 5 hidden nodes, but the training network is one with 10 nodes). The framework of our paper does not have this restriction. Finally, since our paper focus on model interpretability, the framework has been constructed in such a way that all assumptions are minimal/verifiable. This is in contrast to many previous results. For example, Ye and Sun [2018] takes Assumption 6 (which is difficult to check) as given while we can avoid this Assumption using Lemma 3.2.

## 4.2 Future works

There are many avenues for future directions. First, while simulations seem to hint that Group Lasso may not be optimal for feature selection with neural networks, a rigorous answer to this hypothesis requires a deeper understanding of the behavior of the estimator that is out of the scope of this paper. Second, the analyses in this work rely on the fact that the network model is analytic, which puts some restrictions on the type of activation function that can be used. While many tools to study non-analytic networks (as well as other learning settings, e.g., for classification, for regression with a different loss function, or networks that does not have a strict structure of layers) are already in existence, such extensions of the results require non-trivial efforts.

Throughout the paper, we assume that the network is fixed ($p$) when the sample size ($n$) increases. Although this assumption is reasonable in many application settings, in some contexts, for example for the task of identifying genes that increase the risk of a type of cancer, ones would be interested in the case of $p \gg n$. There have been existing results of this type for neural networks from the prediction aspect of the problem Feng and Simon [2017], Farrell et al. [2018] and it would be of general interest how analyses of selection consistency apply in those cases. Finally, since the main interest of this work is theoretical, many aspects of the performance of the GL+AGL across different experimental settings and types of networks are left as subjects of future work.

## Acknowlegments

LSTH was supported by startup funds from Dalhousie University, the Canada Research Chairs program, and the Natural Sciences and Engineering Research Council of Canada (NSERC) Discovery Grant RGPIN-2018-05447. VD was supported by a startup fund from University of Delaware and National Science Foundation grant DMS-1951474.

## Broader Impact

Deep learning has transformed modern science in an unprecedented manner and created a new force for technological developments. However, its black-box nature and the lacking of theoretical justifications have hindered its applications in fields where correct interpretations play an essential role. In many applications, a linear model with a justified confidence interval and a rigorous feature selection procedure is much more favored than a deep learning system that cannot be interpreted. Usage of deep learning in a process that requires transparency such as judicial and public decisions is still completely out of the question.

To the best of our knowledge, this is the first work that establishes feature selection consistency, an important cornerstone of interpretable statistical inference, for deep learning. The results of this work will greatly extend the set of problems to which statistical inference with deep learning can be applied. Medical sciences, public health decisions, and various fields of engineering, which depend upon well-founded estimates of uncertainty, fall naturally on the domain the work tries to explore. Researchers from these fields and the public alike may benefit from such a development and no one is put at disadvantage from this research.

By trying to select a parsimonious and transparent model out of an over-parametrized deep learning system, the approach of this work further provides a systematic way to detect and reduce bias in machine learning analysis. The analytical tools and the theoretical framework derived in this work may also be of independent interest in statistics, machine learning, and other fields of applied sciences.

## Footnotes

[2]The code is available at https://github.com/vucdinh/alg-net.

[3]http://lib.stat.cmu.edu/datasets/boston

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
