[Supplementary Material]

## 5 Appendix

### 5.1 Details of the proof of Theorem 3.4

In the first part of the proof of Theorem 3.4, we established that

$$c_2\|\beta_n - \hat{\alpha}_n\|^\nu \le 2c_1 \frac{\log n}{\sqrt{n}} + \lambda_n C\|\beta_n - \hat{\alpha}_n\|.$$

Using Young's inequality, we have

$$\lambda_n C\|\beta_n - \hat{\alpha}_n\| \le \frac{1}{\nu}\left(\frac{(c_2\nu)^{1/\nu}}{2}\|\beta_n - \hat{\alpha}_n\|\right)^\nu + \frac{\nu-1}{\nu}\left(\frac{2C}{(c_2\nu)^{1/\nu}}\lambda_n\right)^{\nu/(\nu-1)}$$

$$= \frac{c_2}{2}\|\beta_n - \hat{\alpha}_n\|^\nu + \frac{2(\nu-1)C^{\nu/(\nu-1)}}{\nu(c_2\nu)^{1/(\nu-1)}}\lambda_n^{\nu/(\nu-1)}.$$

Combining the two estimates, we have

$$\frac{c_2}{2}\|\beta_n - \hat{\alpha}_n\|^\nu \le 2c_1 \frac{\log n}{\sqrt{n}} + \frac{2(\nu-1)C^{\nu/(\nu-1)}}{\nu(c_2\nu)^{1/(\nu-1)}}\lambda_n^{\nu/(\nu-1)}.$$

### 5.2 Proof of Corollary 3.5

We note that $\phi(\beta_n) \in \mathcal{K}$. Thus,

$$\min_{\alpha \in \mathcal{K}}\|\hat{\alpha}_n - \alpha\| \le \|\hat{\alpha}_n - \phi(\beta_n)\| \le \|\hat{\alpha}_n - \beta_n\| + \|\beta_n - \phi(\beta_n)\|$$

$$\le \|\hat{\alpha}_n - \beta_n\| + \|v_{\beta_n}\|$$

$$\le \|\hat{\alpha}_n - \beta_n\| + \|\hat{v}_n\| + C\|\hat{\alpha}_n - \beta_n\|$$

and the bound can be obtained using the results of Theorem 3.4.

### 5.3 Probabilistic Lipschitzness of the empirical risk

Since both $\mathcal{W}$ and $\mathcal{X}$ are bounded and $f_\alpha$ is analytic, there exist $C_1, C_2 > 0$ such that

$$|\nabla_\alpha f_\alpha(x)| \le C_1 \qquad \text{and} \qquad |f_\alpha(x)| \le C_2 \qquad \forall \alpha \in \mathcal{W}, x \in \mathcal{X}.$$

Therefore,

$$|R(\alpha) - R(\beta)| = \left|\mathbb{E}\left[(Y - f_\alpha(X))^2 - (Y - f_\alpha(X))^2\right]\right|$$

$$\le \mathbb{E}\left|(f_\alpha(X) - f_\beta(X))(2Y - f_\alpha(X) - f_\beta(X))\right|$$

$$\le C_1\|\alpha - \beta\| \cdot \mathbb{E}\left|2Y - f_\alpha(X) - f_\beta(X)\right|$$

$$\le C_1\|\alpha - \beta\| \cdot (2\mathbb{E}\left|Y - f_{\alpha^*}(X)\right| + \mathbb{E}\left|f_\alpha(X) + f_\beta(X) - 2f_{\alpha^*}(X)\right|)$$

$$\le C_1\|\alpha - \beta\|(2\sigma + 4C_2).$$

Similarly,

$$|R_n(\alpha) - R_n(\beta)| \le C_1\|\alpha - \beta\|\left(4C_2 + \frac{2}{n}\sum_{i=1}^n |Y_i - f_{\alpha^*}(X_i)|\right)$$

$$= C_1\|\alpha - \beta\|\left(4C_2 + \frac{2}{n}\sum_{i=1}^n |\epsilon_i|\right).$$

Thus, for all $M_\delta > 4C_1 C_2$,

$$P\left[|R_n(\alpha) - R_n(\beta)| \le M_\delta\|\alpha - \beta\| \,\forall \alpha, \beta \in \mathcal{W}\right]$$

$$\le P\left(\frac{1}{n}\sum_{i=1}^n |\epsilon_i| \le \frac{M_\delta}{2C_1} - 2C_2\right)$$

$$= 1 - P\left(\frac{1}{n}\sum_{i=1}^n |\epsilon_i| \ge \frac{M_\delta}{2C_1} - 2C_2\right)$$

$$\le 1 - \frac{E|\epsilon_1|}{\frac{M_\delta}{2C_1} - 2C_2}.$$

## 5.4 Proof of Lemma 3.3

The proof of this Lemma is similar to that of Lemma 4.2 in Dinh and Ho [2020]. Since the network of our framework is fixed, a standard generalization bound (with constants depending on the dimension of the weight space $\mathcal{W}$) can be obtained. For completeness, we include the proof of Lemma 4.2 in Dinh and Ho [2020] below.

Note that $nR_n(\alpha)/\sigma_e^2$ follows a non-central chi-squared distribution with $n$ degrees of freedom and $f_\alpha(X)$ is bounded. By applying Theorem 7 in Zhang and Zhou (2018) [2], we have

$$\mathbb{P}\left[|R_n(\alpha) - R(\alpha)| > t/2\right]$$
$$\leq 2\exp\left(-\frac{C_1 n^2 t^2}{n + 2\sum_{i=1}^{n}\left[f_\alpha(X) - f_{\alpha^*}(X)\right]^2}\right)$$
$$\leq 2\exp(-C_2 n t^2),$$

for all

$$0 < t < \frac{n + \sum_{i=1}^{n}\left[f_\alpha(X) - f_{\alpha^*}(X)\right]^2}{n}.$$

We define the events

$$\mathcal{A}(\alpha, t) = \{|R_n(\alpha) - R(\alpha)| > t/2\},$$

$$\mathcal{B}(\alpha, t) = \{\exists \alpha' \in \mathcal{W} \text{ such that}$$
$$\|\alpha' - \alpha\| \leq \frac{t}{4M_\delta} \text{ and } |R_n(\alpha') - R(\alpha')| > t\},$$

and

$$\mathcal{C} = \{|R_n(\alpha) - R_n(\alpha')| \leq M_\delta\|\alpha - \alpha'\|, \forall \alpha, \alpha' \in \mathcal{W}\}.$$

Here, $M_\delta$ is defined in Lemma 3.5. By Lemma 3.5, $\mathcal{B}(\alpha, t) \cap \mathcal{C} \subset \mathcal{A}(\alpha, t)$ and $P(\mathcal{C}) \geq 1 - \delta$.

Let $m = dim(\mathcal{W})$, there exist $C_3(m) \geq 1$ and a finite set $\mathcal{H} \subset \mathcal{W}$ such that

$$\mathcal{W} \subset \bigcup_{\alpha \in \mathcal{H}} \mathcal{V}(\alpha, \epsilon) \quad \text{and} \quad |\mathcal{H}| \leq C_3/\epsilon^m$$

where $\epsilon = t/(4M_\delta)$, $\mathcal{V}(\alpha, \epsilon)$ denotes the open ball centered at $\alpha$ with radius $\epsilon$, and $|\mathcal{H}|$ denotes the cardinality of $\mathcal{H}$. By a union bound, we have

$$\mathbb{P}\left[\exists \alpha \in \mathcal{H} : |R_n(\alpha) - R(\alpha)| > t/2\right] \leq 2\frac{C_3(4M_\delta)^m}{t^m}e^{-C_2 n t^2}.$$

Using the fact that $\mathcal{B}(\alpha, t) \cap \mathcal{C} \subset \mathcal{A}(\alpha, t)$, $\forall \alpha \in \mathcal{H}$, we deduce

$$\mathbb{P}\left[\{\exists \alpha \in \mathcal{W} : |R_n(\alpha) - R(\alpha)| > t\} \cap \mathcal{C}\right] \leq C_4 t^{-m}e^{-C_2 n t^2}.$$

Hence,

$$\mathbb{P}\left[\{\exists \alpha \in \mathcal{W} : |R_n(\alpha) - R(\alpha)| > t\}\right] \leq C_4 t^{-m}e^{-C_2 n t^2} + \delta.$$

To complete the proof, we chose $t$ in such a way that $C_4 t^{-m}e^{-C_2 n t^2} \leq \delta$. This can be done by choosing $t = \mathcal{O}(\log n/\sqrt{n})$.

## Footnotes

[2]Zhang, Anru and Yuchen Zhou. "On the non-asymptotic and sharp lower tail bounds of random variables." arXiv preprint arXiv:1810.09006 (2018).