[Reviews · NeurIPS 2020]

Review 1

Summary and Contributions: The paper presents theoretical and empirical results advocating for the adaptive group lasso as a feature selection technique for neural networks.

Strengths: The technique provides a theoretical basis for using the adaptive group lasso to select features in regression environments under moderate assumptions.

Weaknesses: The assumption that the data is positive and completely continuous on it's domain is quite strong, and likely to be broken for most practical datasets. Can the authors comment on the likely behaviour of GL+AGL when this assumption is broken? (Presumably it's broken in Boston Housing, but it's difficult to see any effect given the paucity of results). There is no discussion on the predictive performance of a network trained using this regularisation penalty. The results only show the feature set recovery, but it's conceivable that the net might recover the feature set but predict the output poorly. Without empirical results it's difficult to say. All the neural nets are trained for large numbers of epochs, with no indication on the performance under early stopping, or other techniques to reduce the computational load. Thus it's hard to know if the feature set becomes fixed early in training, or if it's only later in training after the model has fit the training data that the feature set is selected. Without this information (or at least some guidance) it's hard to apply the technique to new problems where it may not be feasible to train for so many epochs. Similarly to the point above, the datasets have at most 50 features. Many real world problems have many more features (especially where feature selection is used), and it's unclear how the technique would scale to those larger problems.

Correctness: I did not verify the proofs in detail. The empirical methodology seems fine, though the experiments have insufficient breadth.

Clarity: Given the technique is gradient based I'd expected some discussion of when features are disabled and if they can be re-enabled (though no gradient would flow to them when they output zero). This can occur in the lasso, and it seems precluded here by the use of SGD.

Relation to Prior Work: Lots of relevant literature is cited, but it's not related in the text to the work at hand, nor is there much discussion of the relationship between this work and other works, beyond the GL+AGL 2020 paper.

Reproducibility: Yes

Additional Feedback: The author response was limited to 1 page, while a 5 page response was submitted, and so to be fair to other submissions we've decided to ignore the response when considering the decision.


Review 2

Summary and Contributions: The paper aims to establish variable selection consistency when an analytic deep neural network model is imposed. In particular, the authors focus on the case when the dimension p is fixed and sample size n goes to infinity. An important assumption made in the paper is the strict positivity of the density over the domain. The authors show that the Adaptive Group Lasso selection procedure is selection-consistent. The general framework of the proof follows that for the analysis of high-dimensional linear models. But the authors also introduce several new technical elements.

Strengths: The authors provide a rigorous characterization of feature selection consistency. Under this definition, with fixed p and strict positive densities, the model identifiability issue can be circumvented. They successfully show variable selection consistency for a rich class of deep neural network models. The results make the deep neural networks model more interpretable.

Weaknesses: My concerns mainly lie in the following two perspectives: 1. Is the model specified in Assumption 2.1 a good one for data generating process? And are the Definition 2.2 and 2.3 appropriate for characterizing feature selection consistency? The authors may argue that these definitions are natural extensions of the concepts for linear models to the deep neural networks case. But my concern is it is well-known that the linear model is quite robust. Even the true date generating process may not be exactly characterized by a linear model, the variable selection results based on a linear model can still help us draw useful conclusions. However, for deep neural networks, the nonlinearity and complicated design may make the model less robust. If the underlying distribution differ from the model in Assumption 2.1, which is probably the case in most applications, then to which level we can trust the variable selection result? 2. To apply the result of Theorem 3.6, given a data set, how should we choose the tuning parameters? Do \gamma and \nu depend on the true model?

Correctness: I did not check the details of the proof. But it seems that the outline of the proof is correct.

Clarity: The paper is well written, and the proofs are also easy to follow. A typo I notice is in line 148, "be setting" should be "by setting".

Relation to Prior Work: The author provide a detailed review of the relevant literature.

Reproducibility: Yes

Additional Feedback:


Review 3

Summary and Contributions: This paper studies the feature selection consistency for a wide class of deep neural network for (adaptive) group Lasso. This paper also shows the advantage of adaptive group lasso over naive group lasso by showing the adaptive group lasso is able to give selection consistency with less regularity condition on the inputs.

Strengths: The main advantage of the paper is that their analysis is applicable to a wide class of neural network. The use of Lojasewicz's inequality to remove the regularity on the Hessian matrix seems new and interesting.

Weaknesses: This paper do not consider the feature selection under high dimensional setting (i.e., the number of feature grows as number of data points grows). And it is unclear that whether the analysis in this paper can be generalized to this case. This paper claims that previous work on feature selection on neural network mainly focus on prediction aspect rather than selection consistency. However, there are many previous works already gives selection consistency analysis. Or the previous prediction result actually implies the feature selection consistency. One key advantage that adaptive group lasso outperforms lasso seems not new and surprising. ****** Updated ****** Thanks authors for the rebuttal. However, I still feel the step from excessive risk to feature selection consistency is not that complicated. And the I believe author should discuss more on the technical advantage of the proposed framework.

Correctness: I believe the correctness of the paper.

Clarity: In general the paper is well written and easy to read.

Relation to Prior Work: The author claims that most previous papers aims at excessive risk bound while this paper focus on selection consistency. However, there are various previous works actually give selection consistency and I believe the author should compare the theoretical analysis with those work.

Reproducibility: Yes

Additional Feedback: Below please find my detailed comments: 1. This paper do not consider the feature selection under high dimensional setting. This paper consider an easier problem setting in which the total number of feature is fixed and does not scale with the number of data points. It is important to study a feature selection method under high dimensional setting, where the number of total feature may grows exponentially fast w.r.t. data points. 2. This paper focus on the feature selection consistency and claim that most previous work mainly focus on excessive error bound because it is hard to address the nonidentifiability issue in neural network. However, there are various previous actually already gives selection consistency result, for example [1,2,3]. And the selection consistency in [4, 5] can be easily obtained with several simple arguments built upon their analysis. I believe it is necessary for authors to do more extensive literature reviews and give more detailed discussion comparing their method and the existing works. [1] Bayesian neural networks for selection of drug sensitive genes [2] Variable Selection via Penalized Neural Network: a Drop-Out-One Loss Approach [3] Posterior Concentration for Sparse Deep Learning [4] Sparse-Input Neural Networks for High-dimensional Nonparametric Regression and Classification [5] Variable Selection with Rigorous Uncertainty Quantification using Deep Bayesian Neural Networks: Posterior Concentration and Bernstein-von Mises Phenomenon 3. I was also wondering about the technical contribution of the proposed analysis framework. How does it make it easier to analyze the deep network compared with the previous method. I think it would be good if the author could discuss the advantage of the proposed framework: what this can do but previous work can not do?


Review 4

Summary and Contributions: This paper presents theoretical results on feature selection consistency for deep neural networks with adaptive group lasso penalty. The authors do so by circumventing the need to account for unidentifiability in deep neural networks, which has been the source of major difficulty in analyzing these methods.

Strengths: This paper generalizes Dinh and Ho (2020) by (i) establishing a way to study the behavior of Group Lasso without a full characterization of the optimal parameter set and (ii) using Lojasewicvz’s inequality to upper-bound the distance between an estimated and an optimal parameters. By doing so, this paper shows that the analyses of feature selection consistency in Dinh and Ho (2020) apply not only to an irreducible feed-forward network with a single hidden layer but also to a broader range of deep analytic neural networks such as feed-forward networks with multiple hidden layers and convolutional neural networks. Also, the empirical illustration of their analyses seems interesting, especially the performance gap between feature selection with AGL + GL and with GL only. In experiments, the authors use deep neural networks to show that their analyses apply to neural networks with multiple hidden layers.

Weaknesses: The experimental results are insufficient. The simulation was done on a single architecture with only 50 input features, and the single real data had only 13 input features. Both the simulated and real data had a small number of input features, which seems inadequate for a study of feature selection problem. Considering that the claim of the paper applies to a broad class of neural networks, the experiments should include those deep neural networks from difference classes.

Correctness: Yes.

Clarity: In general, this paper is easy to follow. Minor commnets: In Definition 2.3 on page 3, I assume n is the number of samples. It’d be good to define this. On line 159, I assume d is the distance between two parameter sets. It’d be good to define this.

Relation to Prior Work: Yes. In Introduction, the authors mention Dinh and Ho (2020) on which this paper seems to be based and clearly stated the difference of this paper and Dinh and Ho (2020).

Reproducibility: Yes

Additional Feedback: ***After author feedback*** Since the authors did not follow the formatting guideline of 1 page for the author feedback, the reviewers were advised not to take into account the author response.

[Author Response · NeurIPS 2020]

We would like to thank the reviewers for their constructive comments. There are some important common comments shared by the reviewers that we want to address before giving specific answers to each reviewer.

## General comments to all reviewers

Some reviewers suggest that the manuscript will benefit for more detailed discussions on the difference between our paper and existing results on the field (both on the findings and the technical aspects). Reviewer 3 has kindly provided some additional references that we would like to integrate into the discussion below. We want to highlight three important points of novelty of our paper:

1. Our paper is the first to establish feature-selection consistency for deep neural networks. This is in contrast to

   - Dinh and Ho [2020], Liang et al. [2018] and Ye and Sun [2018], which only provide results for shallow networks with one hidden layer.
   - Polson and Ročková [2018], Feng and Simon [2017], Liu [2019], which focus on posterior concentration, prediction consistency, parameter-estimation consistency and convergence of feature importance. We note that these convergence properties do not imply feature-selection consistency, and that there's no trivial way to extending these approach to obtain feature-selection-consistency:
     - For classical linear model with lasso estimate, it is known that (see Section 2 of Zou [2006] and the discussion therein) the lasso estimator is parameter-estimation consistent as long as the regularizing parameter $\lambda_n \to 0$ (Zou [2006], Lemmas 2 and 3), but is not feature-selection consistent for $\lambda_n \sim n^{-1/2}$ (Zou [2006], Proposition 1) or for all choices of $\lambda_n$ if some necessary condition on the covariance matrix is not satisfied (Zou [2006], Theorem 1).
       For both linear model and neural network, parameter-estimation consistency directly implies prediction consistency and convergence of feature importance, so in general, it's not possible to obtain feature-selection consistency just based on those convergence properties. Posterior concentration, which studies the contraction of the posterior distribution around the set of risk minimizers, is also a convergence property that is regarded as comparable with parameter-estimation consistency, but it's unlikely that the property is as strong as feature-selection consistency.
     - Ye and Sun [2018], which is built upon Feng and Simon [2017], can only achieve feature-selection consistency for networks with one hidden layer, while Feng and Simon [2017] can achieve parameter-estimation consistency (and thus, convergence of feature importance) for deep networks. This highlights the difficulty of achieving feature-selection-consistency for deep networks, and that it is not trivial to obtain such a result from Feng and Simon [2017] and Liu [2019].

2. Pre-existing works usually assume that the network used for training has exactly the same size as the minimal network that generates the data. This is assumption is either made explicitly (as in Dinh and Ho [2020]) or implicitly implied by the regularity of the Hessian matrix at the optima (Assumption 5 in Ye and Sun [2018] and Condition 1 in Feng and Simon [2017]). We note that these latter two conditions cannot be satisfied if the size of the training network is not "correct" (for example, when data is generated by a one-hidden-layer network with 5 hidden nodes, but the training network is one with 10 nodes). The framework of our paper does not have this restriction (the use of Lojasewicz's inequality solves this problem)

3. Since our paper focus on model interpretability, the framework has been constructed in such a way that all assumptions are minimal/verifiable. This is in contrast to many previous results. For example, Ye and Sun [2018] takes Assumption 6 (which is very difficult to check) as given while we can avoid this Assumption using Lemma 3.2.

Secondly, some reviewers commented that our simulations are insufficient in illustrating the performance of the proposed method. We would like to clarify that this work is intended to be a theory paper and not a proof-by-simulation methodology one. In other words, the existence of the simulation part is solely to highlight the theoretical findings and visualize the performance gap between GL and GL+AGL in simple settings. We believe the current simulation coherently established these points.

Specific comments to the reviewers are as follows.

## Comments to Reviewer 1

*The assumption that the data is positive and completely continuous on it's domain is quite strong, and likely to be broken for most practical datasets. Can the authors comment on the likely behaviour of GL+AGL when this assumption is broken? (Presumably it's broken in Boston Housing, but it's difficult to see any effect given the paucity of results).*

≫ First, we want to clarify that the assumption we made in the paper is that the *density function* of the data is absolutely continuous and positive on its domain (not the data). It is correct that our analysis focus on the case when $\mathcal{X}$ is an open domain, and thus the variables are assumed to be continuous. However, we note that the result can be straightforward to accommodate discrete variables with a few adjustments.

*There is no discussion on the predictive performance of a network trained using this regularization penalty. The results only show the feature set recovery, but it's conceivable that the net might recover the feature set but predict the output poorly. Without empirical results it's difficult to say.*

≫ We want to emphasize while standard machine learning usually regards feature selection as a mean to improve prediction accuracy, in many application contexts, the goal is to identify which features affect the outcomes rather than to make the best prediction (e.g., identifying key genes that affect the survival rates of cancer patients is more important than predicting the survival rates using the whole genomes). In this work, our main focus is to show that a feature-selection-consistent method can be achieved for the neural network model and our simulations illustrate the performance of the approach in this aspect.

*All the neural nets are trained for large numbers of epochs, with no indication on the performance under early stopping, or other techniques to reduce the computational load. Thus it's hard to know if the feature set becomes fixed early in training, or if it's only later in training after the model has fit the training data that the feature set is selected. Without this information (or at least some guidance) it's hard to apply the technique to new problems where it may not be feasible to train for so many epochs.*

≫ We agree with the Reviewer that guidance on apply the method along with other techniques to reduce the computational load is an important question that we would like to address in the future. We note that feature selection (or interpreting a model) is a difficult task that probably requires high accuracy in training, which may require more computational efforts than usual.

*Similarly to the point above, the datasets have at most 50 features. Many real world problems have many more features (especially where feature selection is used), and it's unclear how the technique would scale to those larger problems.*

≫ As mentioned in the general comments section, the focus of our work is theoretical performance and the simulation is only for illustration. We note that our method can be easily adapted from weighted lasso,

and only double the computational cost of a standard prediction problem with lasso. From that aspect, we expect that our approach would scale well to larger problems in terms of computational costs.

*Given the technique is gradient based I'd expected some discussion of when features are disabled and if they can be re-enabled (though no gradient would flow to them when they output zero). This can occur in the lasso, and it seems precluded here by the use of SGD.*

⋙ We are not quite sure that we understand your comment correctly. Our proximal algorithm continuously soft-threshold the parameter to zero, but never actually disable any of the features. We also note that unlike network with ReLU activations that sometimes lead to dead nodes with no gradient flow, our network is analytic and if an input node is significant, then the gradient to the parameter associated with it will not be killed completely.

## Comments to Reviewer 2

*Is the model specified in Assumption 2.1 a good one for data generating process? And are the Definition 2.2 and 2.3 appropriate for characterizing feature selection consistency? The authors may argue that these definitions are natural extensions of the concepts for linear models to the deep neural networks case. But my concern is it is well-known that the linear model is quite robust. Even the true date generating process may not be exactly characterized by a linear model, the variable selection results based on a linear model can still help us draw useful conclusions. However, for deep neural networks, the nonlinearity and complicated design may make the model less robust. If the underlying distribution differ from the model in Assumption 2.1, which is probably the case in most applications, then to which level we can trust the variable selection result?*

⋙ We note that Assumption 2.1 is a common assumption for this framework. One alternative approach (as in Ye and Sun [2018], Feng and Simon [2017]) for this is to define:

$$EQ^* = \arg \min_{\alpha} R(y, f_\alpha(x))$$

and assume that for all $\alpha \in EQ^*$, all weights tied to the insignificant features are zero (which is compatible with Definition 2.2 and 2.3). The results of our paper can be straightforwardly extended to this setting with minimal changes. However, while this avoids the exact model-based assumption, this comes with the cost of losing verifiability/interpretability of the assumption, and may not address your concern directly. We believe that the robustness of feature selection with deep learning is an open problem.

*To apply the result of Theorem 3.6, given a data set, how should we choose the tuning parameters? Do $\gamma$ and $\nu$ depend on the true model?*

⋙ In practice, for a fixed dataset, all tuning parameters are chosen using cross-validation. In Theorem 3.6, $\gamma$ can be any positive number and do not depend on the model. In the simulation, $\gamma = 2$ but any positive number works. $\nu$ does depend on the true model but it's not a tuning parameter.

## Further comments to Reviewer 3

1. *This paper do not consider the feature selection under high dimensional setting. This paper consider an easier problem setting in which the total number of feature is fixed and does not scale with the number of data points. It is important to study a feature selection method under high dimensional setting, where the number of total feature may grows exponentially fast w.r.t. data points.*

2. *This paper focus on the feature selection consistency and claim that most previous work mainly focus on excessive error bound because it is hard to address the nonidentifiability issue in neural network.*

*However, there are various previous actually already gives selection consistency result, for example [1,2,3]. And the selection consistency in [4, 5] can be easily obtained with several simple arguments built upon their analysis. I believe it is necessary for authors to do more extensive literature reviews and give more detailed discussion comparing their method and the existing works.*

*[1] Bayesian neural networks for selection of drug sensitive genes*

*[2] Variable Selection via Penalized Neural Network: a Drop-Out-One Loss Approach*

*[3] Posterior Concentration for Sparse Deep Learning*

*[4] Sparse-Input Neural Networks for High-dimensional Nonparametric Regression and Classification*

*[5] Variable Selection with Rigorous Uncertainty Quantification using Deep Bayesian Neural Networks: Posterior Concentration and Bernstein-von Mises Phenomenon*

3. *I was also wondering about the technical contribution of the proposed analysis framework. How does it make it easier to analyze the deep network compared with the previous method. I think it would be good if the author could discuss the advantage of the proposed framework: what this can do but previous work can not do?*

⋙ As we mentioned in the general comment, although we focus on low-dimensional settings, our problem is not trivial because (1) we consider deep networks and, (2) we focus on feature-selection consistency.

We recall that feature-selection consistency is a very strong property, which states that with finite data, the estimated weights that correspond to the insignificant features are *exactly* zero. This is in contrast to other properties, such as prediction consistency, parameter-estimation consistency, and convergence of feature importance, which hold if the estimated weights that correspond to the insignificant features *converges* (but not necessarily equal to) zero.

As you have pointed out, there are some previous work about feature selection consistency, such as Dinh and Ho [2020], Liang et al. [2018] and Ye and Sun [2018]. However, their results can only apply to shallow networks with one hidden layer. All of pre-existing works for deep networks (Polson and Ročková [2018], Feng and Simon [2017], Liu [2019]) are only about prediction consistency, parameter-estimation consistency and convergence of feature importance. Extending these work to feature-selection-consistency is non-trivial. For example, Ye and Sun [2018], which is built upon Feng and Simon [2017], can only achieve feature-selection consistency for networks with one hidden layer, while Feng and Simon [2017] can achieve parameter-estimation consistency (and thus, convergence of feature importance) for deep networks.

Another strong point of our work is that the conditions imposed on the framework are minimal/verifiable and can work well in cases when the size of the generating network is not known exactly.

We agree that studying feature selection under high dimensional setting is an important problem that is even more challenging for deep networks. We hope to address this in future work.

## Comments to Reviewer 4

*The experimental results are insufficient. The simulation was done on a single architecture with only 50 input features, and the single real data had only 13 input features. Both the simulated and real data had a small number of input features, which seems inadequate for a study of feature selection problem. Considering that the claim of the paper applies to a broad class of neural networks, the experiments should include those deep neural networks from difference classes.*

⋙ As mentioned in the general comments section, the focus of our work is theoretical performance and the simulation is only for illustration. We think the approach can be further extended/optimized in

theory, algorithmic variants, and experimentations with larger and different types of networks; however, we envisioned that these would be parts of future works.

*Minor commnets:*
*In Definition 2.3 on page 3, I assume n is the number of samples. It'd be good to define this.*
*On line 159, I assume d is the distance between two parameter sets. It'd be good to define this.*

⋙ Thank you for your comment. All minor typos will be fixed in the vision.

# References

Vu Dinh and Lam Ho. Consistent feature selection for neural networks via Adaptive Group Lasso. *arXiv preprint arXiv:2006.00334*, 2020.

Jean Feng and Noah Simon. Sparse-input neural networks for high-dimensional nonparametric regression and classification. *arXiv preprint arXiv:1711.07592*, 2017.

Faming Liang, Qizhai Li, and Lei Zhou. Bayesian neural networks for selection of drug sensitive genes. *Journal of the American Statistical Association*, 113(523):955–972, 2018.

Jeremiah Zhe Liu. Variable selection with rigorous uncertainty quantification using deep bayesian neural networks: Posterior concentration and bernstein-von mises phenomenon. *arXiv preprint arXiv:1912.01189*, 2019.

Nicholas G Polson and Veronika Ročková. Posterior concentration for sparse deep learning. In *Advances in Neural Information Processing Systems*, pages 930–941, 2018.

Mao Ye and Yan Sun. Variable selection via penalized neural network: a drop-out-one loss approach. In *International Conference on Machine Learning*, pages 5620–5629, 2018.

Hui Zou. The adaptive lasso and its oracle properties. *Journal of the American statistical association*, 101 (476):1418–1429, 2006.



[Meta-Review · NeurIPS 2020]

This paper shows that the adaptive group lasso feature selection method, more specifically a combined strategy called GL+AGL, is selection consistent for a very general class of Deep Neural Networks, provided the the DNN interacts with the input through a finite set of linear units. This is an important property since it provides a guarantee that, with enough training examples, GL+AGL will effectively identify the set of relevant inputs; making the DNN more interpretable. The general structure of the proof follows the analysis of high-dimensional linear models, but new technical elements are introduced to tackle de difficulties introduced when the linear transformation of the first layer is followed by a sequence of non-linear transformations typically used in DNNs. Finally the numerical experiments provide evidence that the popular group lasso method might be an inefficient feature selection method for DNNs.